# Endopiriform neurons projecting to ventral CA1 are a critical node for recognition memory

**Naoki Yamawaki[1,2,3], Hande Login[1,2,3], Solbjørg Østergaard Feld-Jakobsen[1], Bernadett Mercedesz Molnar[1], Mads Zippor Kirkegaard[1], Maria Moltesen[1], Aleksandra Okrasa[1], Jelena Radulovic[1,2,3,4,5], Asami Tanimura[1,2,3]***

[1]Department of Biomedicine, Aarhus University, Aarhus, Denmark; [2]PROMEMO, The Center for Proteins in Memory, Aarhus University, Aarhus, Denmark; [3]DANDRITE, The Danish Research Institute of Translational Neuroscience, Aarhus University, Aarhus, Denmark; [4]Dominick P. Purpura Department of Neuroscience, Albert Einstein College of Medicine, New York, United States; [5]Department of Psychiatry and Behavioral Sciences, Albert Einstein College of Medicine, New York, United States

## eLife Assessment

This **important** study offers insights into the function and connectivity patterns of a relatively unknown afferent input from the endopiriform to the CA1 subfield of the ventral hippocampus, suggesting a neural mechanism that suppresses the processing of familiar stimuli in favor of detecting memory guided novelty. The strength of evidence is **convincing**, with careful anatomical and electrophysiological circuit characterization. The work will be of broad interest to researchers studying the neural circuitry of behavior.

**\*For correspondence:**
asami.tanimura@biomed.au.dk

**Competing interest:** The authors declare that no competing interests exist.

## Abstract

The claustrum complex is viewed as fundamental for higher-order cognition; however, the circuit organization and function of its neuroanatomical subregions are not well understood. We demonstrated that some of the key roles of the CLA complex can be attributed to the connectivity and function of a small group of neurons in its ventral subregion, the endopiriform (EN). We identified a subpopulation of EN neurons by their projection to the ventral CA1 ($EN^{vCA1-proj.}$ neurons), embedded in recurrent circuits with other EN neurons and the piriform cortex. Although the $EN^{vCA1-proj.}$ neuron activity was biased toward novelty across stimulus categories, their chemogenetic inhibition selectively disrupted the memory-guided but not innate responses of mice to novelty. Based on our functional connectivity analysis, we suggest that $EN^{vCA1-proj.}$ neurons serve as an essential node for recognition memory through recurrent circuits mediating sustained attention to novelty, and through feed-forward inhibition of distal vCA1 neurons shifting memory-guided behavior from familiarity to novelty.

## Introduction

The claustrum (CLA) complex, an evolutionarily conserved brain region found across mammalian species, reptiles, and birds, is hypothesized to serve as a node for establishing higher-order cognitive functions by coordinating neuronal activity on a global scale (*Narikiyo et al., 2020*; *Norimoto et al., 2020*; *Crick and Koch, 2005*). This view is based on its extensive connections with many cortical and subcortical areas (*Smith et al., 2020*; *Goll et al., 2015*). Some of these functions include sensory perception and attention, which may affect memory processing, including working memory,

associative memory, and recognition memory (*Gardiner and Parkin, 1990*; *Siegel and Castel, 2018*; *Oberauer, 2019*). Consistent with this view, abnormalities of the CLA complex is found in major cognitive disorders, including Alzheimer's disease, schizophrenia, and attention-deficit/hyperactivity disorders (*Chen et al., 2023*; *Cascella et al., 2011*; *Wang et al., 2013*). However, microcircuitry and functional segregation of the individual constituents of the CLA complex have remained unclear.

The cytoarchitecture and genetic expression broadly divide the CLA complex into dorsal and ventral parts, with the dorsal part typically referred to as 'CLA' and the ventral part having its own nomenclature: 'EN' in rodents (*Smith et al., 2019*; *Grimstvedt et al., 2023*). Current evidence indicates that the CLA forms reciprocal connections with dorsal cortices containing sensory, motor, and association areas generating diverse physiological effects depending on the cortical targets and their activity patterns (*Smith et al., 2020*; *Goll et al., 2015*). In contrast, EN primarily provides inputs to the piriform cortex and limbic systems (*Behan and Haberly, 1999*; *Majak and Moryś, 2007*), suggesting its distinct circuit organization and role in bridging olfactory information processed by the piriform cortex and memory processed by the limbic systems (*Fu et al., 2004*; *Watson et al., 2017*).

One of the distinct targets of EN, the ventral CA1 (vCA1), is known to coordinate in a range of behaviors related to exploration and recognition memory, and these functions are thought to be regulated by its afferent inputs in a domain-specific manner (*Pi et al., 2020*; *Ciocchi et al., 2015*; *Hunsaker et al., 2008*; *Okuyama et al., 2016*; *Kesner et al., 2011*; *Meira et al., 2018*; *Tanimizu et al., 2017*; *Titulaer et al., 2021*; *Broadbent et al., 2010*). We, therefore, hypothesized that EN is a key node for the establishment of recognition memory. To address this, we set out to characterize the circuit and function of EN neurons defined by their projection to vCA1 (EN [vCA1-proj.] neurons) using genetic tools and mouse models of social and object recognition memory.

We found EN [vCA1-proj.] neurons innervated multiple components of the limbic system except the amygdala and prefrontal cortex and produced potent feedforward inhibitory control over vCA1 pyramidal neurons. During the recognition memory test, the activity of EN [vCA1-proj.] neurons were condensed around conspecifics or objects where mice spent most time on. However, disruption of EN [vCA1-proj.] activity only impaired memory-guided exploration of novel stimuli without affecting innate exploration induced by novelty. These findings demonstrate that EN subserves some of the key functions required for recognition memory governed by specific limbic systems.

## Results

### Endopiriform represents a major afferent of the ventral CA1

To assess the significance of EN as a vCA1 afferent, we injected a retrograde tracer (60 nl) into the vCA1 of mice and compared the number of the retrogradely labeled neurons in the EN and other brain regions (*Figure 1A and B* and *Figure 1—figure supplement 1A*, **Methods**). We found labeled neurons in multiple areas including the septum, entorhinal cortex, and basolateral amygdala (*Figure 1C–F* and *Figure 1—figure supplement 1B*; *Tao et al., 2021*; *Gergues et al., 2020*). In addition to these well-known vCA1 afferents, prominent labeling was observed in the area defined as EN in Allen Brain Atlas (*Figure 1G*, **Methods**). The number of presynaptic neurons in EN was ~10- fold lower than the entorhinal cortex (data not shown). However, the normalized count of presynaptic neurons in EN was significantly greater than the basolateral amygdala, lateral septum, and medial septum (*Figure 1H*).

The vCA1-projecting neurons in the EN were spread along an antero-posterior axis with two peak densities, one in anterior and another in posterior to the bregma (*Figure 1I and J*). Number of labeled neurons was greater in posterior EN (*Figure 1J*). In contrast, the same analysis along the dorso-ventral axis showed a single dominant peak at the depth consistent with the location of EN (*Figure 1—figure supplement 1C*).

To further confirm the location of these vCA1-projecting neurons, we injected a retrograde tracer (60 nl for each) in one color into vCA1 and a retrograde tracer in another color into a cortical area known to receive projections from EN or 'CLA' representing a dorsal part of CLA complex (*Figure 2A*). We targeted the prefrontal cortex to label EN, and the anterior cingulate cortex, the motor cortex, and the dorsomedial entorhinal cortex to label CLA (*Figure 2C–F*; *Kitanishi and Matsuo, 2017*; *Qadir et al., 2018*; *Jackson et al., 2018*; *Smith and Alloway, 2010*).

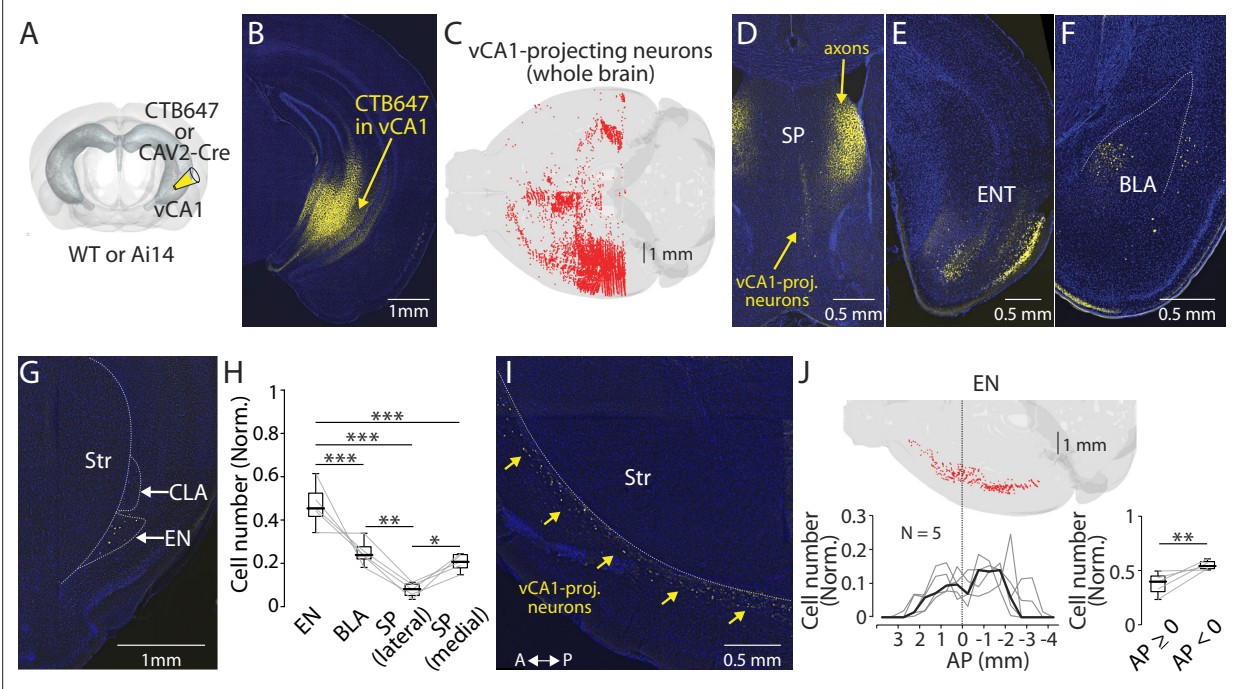

**Figure 1.** Endopiriform projects to the ventral CA1. (**A**) Schematic diagram of a retrograde tracer injection performed. CAV-2-Cre or CTB647 was used in combination with Ai14 or wild-type mice, respectively. (**B**) An example fluorescent image of the injection site. (**C**) Distribution of retrogradely labeled ventral CA1 (vCA1)-projecting neurons in the entire brain as a result of injection in panel B. (**D–G**) Fluorescent images showing retrogradely labeled vCA1-projecitng neurons in the septum (SP), entorhinal cortex (ENT), basolateral amygdala (BLA), and endopiriform (EN). Str: Striatum, CLA: Claustrum. (**H**) Box plots showing a relative number of vCA1-projecting neurons in different brain areas. Cell numbers in each brain area were normalized by summed cell numbers of EN, BLA, SP (lateral), and SP (medial). Cell numbers in each brain areas of individual brains are presented in *Table 1*. Median values for EN, BLA, SP (lateral), and SP (medial) is 0.45, 0.24, 0.08, and 0.21, respectively. Statistics: EN vs BLA, SP (lateral), or SP (medial), p<0.001 for all; BLA vs SP (lateral) or SP (medial): p=0.002 and 0.694, respectively; SP (lateral) vs SP (medial) p=0.023. One-way ANOVA, repeated comparison with Tukey-Kramer test. N=5. (**I**) An example fluorescent image of the horizontal section showing retrogradely labeled vCA1-projecting neurons in EN. A: anterior, P: posterior. (**J**) Top: Example map showing the distribution of vCA1-projecting neurons in the entire EN in one mouse. Bottom left: A plot showing a quantified distribution at a bin size of 1 mm. Each bin was normalized to the total number of labeled neurons in EN. Data from each mouse is shown in gray. Median is indicated in black. Bottom right: Box plots comparing a number of vCA1-projecting neurons in anterior and posterior EN. Data were normalized to the total number of labeled neurons in EN. Median value for anterior vs posterior was 0.39 vs 0.54. Statistics: p=0.008, Rank-sum.

The online version of this article includes the following source data and figure supplement(s) for figure 1:

**Source data 1.** Presynaptic cell numbers projecting to vCA1.

**Source data 2.** Presynaptic cell numbers projecting to vCA1.

**Figure supplement 1.** Whole brain distribution of ventral CA1 (vCA1)-projecting neurons.

We found there is a major spatial overlap in labeling of vCA1-projecting neurons with prefrontal cortex-projecting neurons, but not with other cortical projection neurons (*Figure 2G–J*). Moreover, around 30% of vCA1-projecting neurons were found to be double-labeled when a cortical injection was made into the prefrontal cortex, suggesting some vCA1-projecting neurons send axons to this cortical area (*Figure 2K and L*).

**Table 1.** Presynaptic cell numbers projecting to vCA1 in different brain areas.

| Brain areas | Brain #1 | Brain #2 | Brain #3 | Brain #4 | Brain #5 |
|---|---|---|---|---|---|
| EN | 288 | 201 | 310 | 210 | 250 |
| BLA | 85 | 107 | 175 | 208 | 118 |
| SP (lateral) | 27 | 36 | 69 | 69 | 17 |
| SP (medial) | 69 | 111 | 130 | 127 | 121 |
| Sum | 469 | 455 | 684 | 614 | 506 |

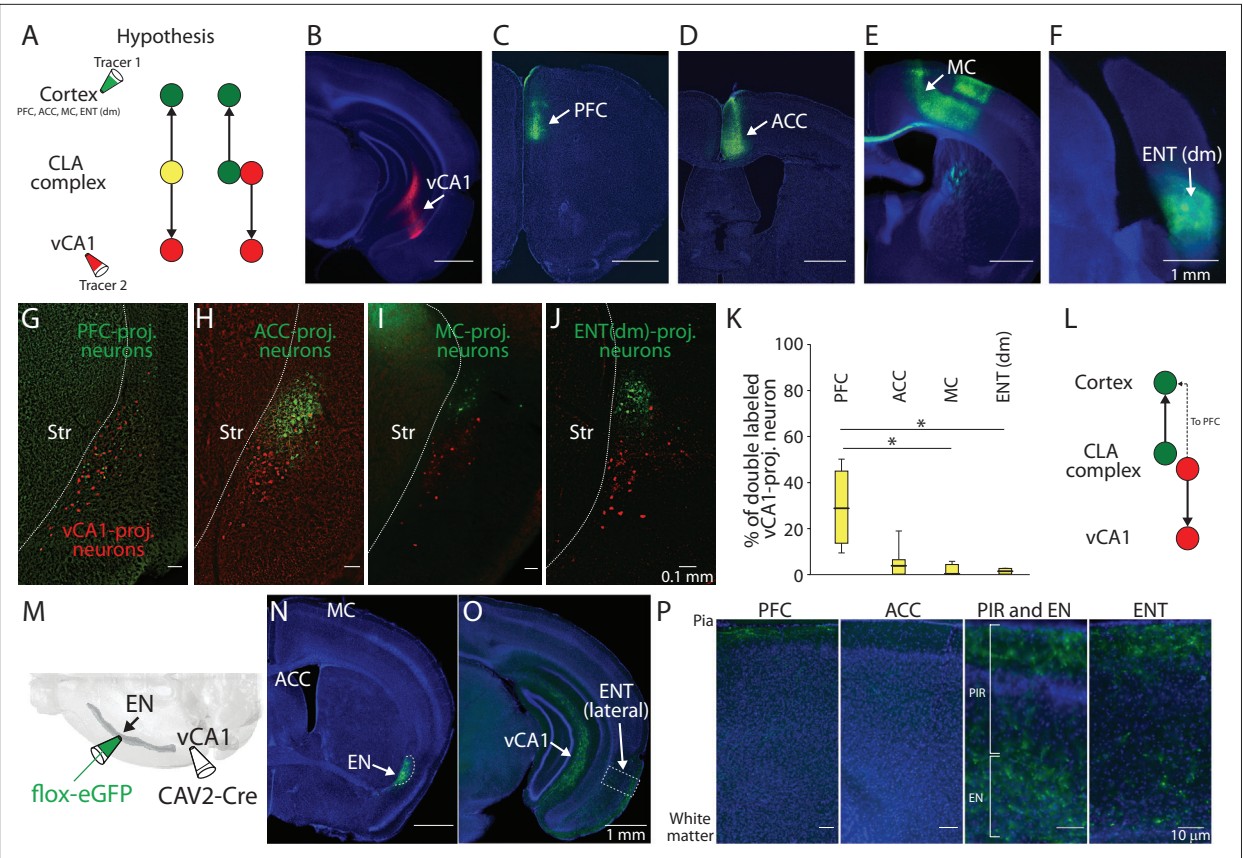

**Figure 2.** Axon branching of ventral CA1 (vCA1)-projecting endopiriform (EN) neurons. (**A**) Schematic diagram of injection performed and two hypothesized outcomes. Yellow color indicates double-labeled neurons. PFC: Prefrontal cortex, ACC: Anterior cingulate cortex, MC: Motor cortex, ENT(dm): Dorsomedial entorhinal cortex, CLA: claustrum (**B–F**) Example fluorescent images of the injection site for each region indicated. (**G–J**) Example fluorescent images of retrogradely labeled neurons in CLA complex. vCA1-projecting neurons were colored in red and cortex-projecting neurons were colored in green. (**K**) Box plots showing the percentage of double-labeled vCA1-projecting neurons for each cortical injection. Medial values for PFC, ACC, MC, and ENT(dm) were 28.6, 4.6, 0, and 1.2, respectively. Statistics: PFC vs ACC, MC or ENT(dm), p=0.405, 0.015, and 0.047, respectively. ACC vs MC or ENT (dm), p=0.422 and 0.64, respectively. MC vs ENT (dm), p=0.993. One-way ANOVA with post-hoc Kruskal-Wallis test. Slice and animal numbers: 16 and 4 for PFC, 20 and 5 for ACC, 10 and 3 for MC, and 18 and 4 for ENT(dm). (**L**) Schematic diagram of projection pattern found in the experiment shown in A. (**M**) Schematic diagram of the intersectional approach performed to label vCA1-projecting EN neurons. (**N–O**) Example fluorescent images showing vCA1-projecting axons in different brain areas. All images were from the same mouse. (**P**) Magnified images showing vCA1-projecting axons in cortical layers. Magnified areas are indicated in (**O**) and fig. **S2A-C** as dashed squares. PIR: Piriform cortex.

The online version of this article includes the following source data and figure supplement(s) for figure 2:

**Source data 1.** Single and double labeled EN cell numbers.

**Figure supplement 1.** Axon branching of ventral CA1 (vCA1)-projecting endopiriform (EN) neurons.

To further determine the cortical and subcortical innervation pattern of vCA1-projecting EN neurons, we used an intersectional approach to label their somata and axons with eGFP. CAV2-Cre (60 nl) was injected into the vCA1 and flox-eGFP (100 nl) was injected into the EN (*Figure 2M*). Apart from vCA1, axons from vCA1-projecting neurons were found in various cortical areas including the prefrontal cortex, lateral entorhinal cortex, and the piriform cortex (*Figure 2N–P* and *Figure 2—figure supplement 1A-C*). Projection to the prefrontal cortex was much sparser relative to that of vCA1, as expected based on the retrograde labeling data (*Figure 2K*). The connectivity of vCA1-projecting EN neurons to the amygdala, which represents the major component of limbic systems, was also sparse relative to vCA1 (*Figure 2—figure supplement 1C*), indicating vCA1 is the main target of these EN neurons in this system. Within the hippocampus, very sparse EN axons were observed in vCA3 and dentate gyrus (*Figure 2—figure supplement 1D*).

Taken together, these findings indicate EN represents a major afferent of vCA1. Moreover, EN neurons projecting their axons to vCA1 also send strong collaterals to the lateral entorhinal cortex

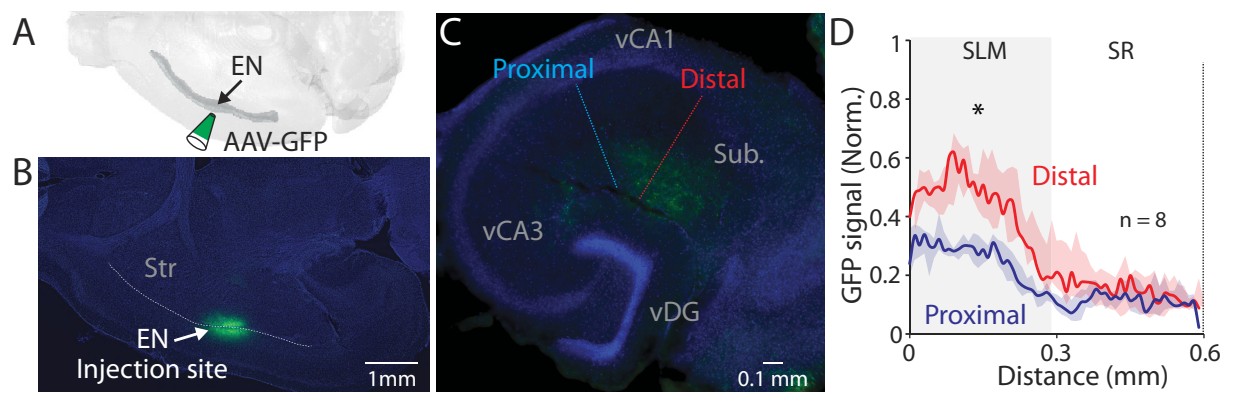

**Figure 3.** EN projection pattern in ventral CA1 (vCA1). (**A**) Schematic diagram of injection performed. (**B**) An example fluorescent image of a horizontal brain section containing an injection site in endopiriform (EN). (**C**) An example fluorescent image of the ventral hippocampal slice with EN axons resulted from the injection in panel **B**. Dashed lines indicate the distal and proximal vCA1 regions where GFP signals were measured for the plot in panel **D**. (**D**) Normalized GFP signals across the layer. All signals were normalized by the maximum GFP signal in the distal subregion. Median of the area under the curve: distal; 11.12, proximal; 5.99. distal vs proximal, p=0.016. Wilcoxon signed-rank test. n=8 slices (N=4 mice). SR: stratum radiatum, Py: pyramidal layer.

The online version of this article includes the following source data and figure supplement(s) for figure 3:

**Source data 1.** Normalized GFP signals in hippocampal slices.

**Figure supplement 1.** Projections of EN^vCA-proj. neurons to the hippocampus.

and piriform cortex, but relatively weak to the prefrontal cortex or other limbic structures. Since our subsequent study focused on EN neurons defined by their projection to vCA1, we refer to them as EN^vCA1-proj. neurons.

## Projection pattern and synaptic connectivity of EN axons in vCA1

We next investigated the projection pattern of EN axons in the hippocampus by injecting an anterograde tracer (AAV-GFP, 60 nl) into the EN (*Figure 3A and B*). Analysis of different parts of the hippocampal section indicated EN axons were mostly confined in the SLM layer of the distal part of vCA1 (*Figure 3C and D*). Projection to intermediate or dorsal CA1 was limited or undetectable (*Figure 3— figure supplement 1A-C*).

Since SLM mainly consists of GABAergic neurons (*Pelkey et al., 2017*), we hypothesized that EN axons form synapses with this cell type. To test this, we first applied the monosynaptic rabies tracing technique to Vgat-Cre mice (*Figure 4A*, **Methods**). The distribution of starter cells (identified by GFP and mCherry co-labeling) was confirmed to be within the ventral-intermediate region of the hippocampus using AMaSiNe (*Figure 4B* and *Figure 4—figure supplement 1*; *Song et al., 2020*). In these mice, the presynaptic neurons were consistently observed in EN in addition to other expected areas (e.g. septum and entorhinal cortex) (*Figure 4C and D*, *Figure 4—figure supplement 1B-F*; *Freund and Antal, 1988*; *Bilash et al., 2023*). The density of the presynaptic neurons in EN was greater at the posterior region to the bregma (*Figure 4D*), consistent with our earlier observation (*Figure 1J*).

To determine how EN inputs are integrated into the local circuit in vCA1, we systematically assessed the connections between EN axons and GABAergic neurons in different layers of distal vCA1. We expressed channelrhodopsin-2 (ChR2) into the EN axons by injecting AAV-ChR2-mCherry (100 nl) into EN and performed whole-cell recording from GABAergic neurons in acute vCA1 slices in the presence of TTX (1 µM) and 4-AP (100 µM) to isolate monosynaptic connection (*Petreanu et al., 2009*). To ensure the recording from GABAergic neurons, we used Vgat::Ai14 mice in which all GABAergic neurons are labeled with tdTomato (*Figure 4E–G*). We recorded from 3 to 4 GABAergic neurons located in different laminar positions in distal vCA1 (Py-SR border, SR, SR-SLM border, or SLM) in the same slice to construct a 'laminar profile' of EN→vCA1 inputs (*Figure 4F and G*). Whole-field LED stimulation evoked excitatory postsynaptic current (EPSC) in most GABAergic neurons recorded. However, contrary to the expectation, connection probability and strength were lowest in neurons in SLM compared to neurons in other layers (*Figure 4H–J*).

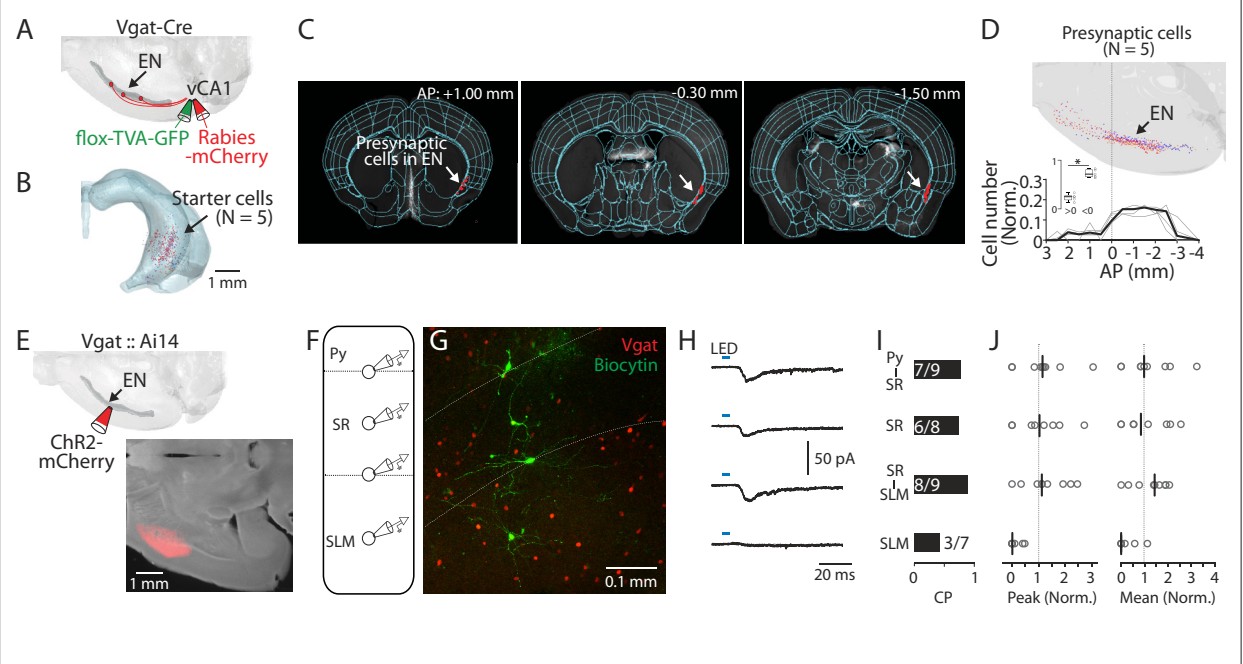

**Figure 4.** EN[vCA1-proj]. neurons target ventral CA1 (vCA1) interneurons. (**A**) Schematic diagram of injection performed for monosynaptic rabies tracing. Flox-TVA-GFP (60 nl) was injected into the vCA1. 4 wk later, Rabies-mCherry (60 nl) was injected. (**B**) Location of starter cells in the hippocampus. Data from five brains (each brain represented by a different color) were overlaid. (**C**) Example images showing presynaptic cells in endopiriform (EN) (white arrow). (**D**) Top: Location of presynaptic EN neurons. Data from five brains were overlaid (different brains were coded by different colors). Bottom: Plots indicate the distribution of presynaptic cells in EN along the AP axis. Cell counts in a given bin (1 mm) were normalized by the total cell number in EN. Thin gray lines indicate individual data. A thick black line indicates median. Inset showing plot comparing cell number in anterior and posterior location as in *Figure 1J*. Median: anterior vs posterior, 0.25 vs 0.72, p=0.031; Wilcoxon signed-rank test. (**E**) Top: Schematic diagram of the injection performed. Bottom: An example fluorescent image of injection site in EN. (**F**) Schematic diagram of recordings performed. Recordings were performed from identified GABAergic neurons sequentially in a random order. (**G**) An example confocal image of an acute vCA1 slice is used for recordings. Recorded GABAergic neurons were filled with biocytin and post-labeled with Alexa488. White dashed lines indicate the border of Py-SR and SR-SLM. (**H**) Median traces of photo-evoked EPSCs were recorded from neurons at different laminar positions. (**I**) Connection probability (CP) of EN axons and recorded neurons at different laminar positions. Numbers in bars indicate number of response-positive neurons /total recorded neurons. (**J**) Left: Peak amplitudes of evoked EPSCs (normalized to total input from recorded slice). Right: the same as left but for mean amplitude. Gray circles indicate each neuron. Black lines indicate the median. Median (peak amp): Py-SR vs SR, SR-SLM or SLM: 1.1 vs 1.0 vs 1.1 vs 0.0. [Py-SR vs SR, SR/SLM vs SLM], p=0.999, 0.988, 0.076; [SR vs SR-SLM or SLM], p=0.972, 0.111; [SR-SLM vs SLM], p=0.039; one-way ANOVA, repeated comparison with Tukey-Kramer test. Median (mean amp): Py-SR vs SR, SR-SLM, or SLM: 1.0 vs 0.8 vs 1.4 vs 0.0. [Py-SR vs SR, SR-SLM or SLM], p=0.991, 0.999, and 0.141; [SR vs SR-SLM or SLM], p=0.978, and 0.251; [SR-SLM vs SLM], p=0.114; one-way ANOVA, repeated comparison with Tukey-Kramer test. N=5 mice.

The online version of this article includes the following source data and figure supplement(s) for figure 4:

**Source data 1.** Number of presynaptic cells projecting to ventral hippocampal interneurons.

**Source data 2.** Recorded EN-evoked EPSPs from ventral CA1 interneurons.

**Figure supplement 1.** Presynaptic neurons of ventral hippocampal GABAergic neurons.

Taken together, these results indicate EN axons preferentially innervate the GABAergic neurons spanning across the Py-SR border to the SR-SLM border in distal vCA1.

## EN axons produce feedforward inhibition onto vCA1 pyramidal neurons

SLM also contains the apical tuft dendrite of pyramidal neurons, which could be a target of EN axons in addition to GABAergic neurons. To test this, we expressed ChR2 in EN axons but made whole-cell recording from pyramidal neurons in distal vCA1 in the presence of TTX and 4-AP. Since CA1 pyramidal neurons can be differentiated into superficial and deep-layer neurons with distinct anatomy and function (*Danielson et al., 2016*), we recorded from neurons in each layer as well as nearby GABAergic neurons at the Py-SR and SR-SLM borders for the comparison (*Figure 5A*). We found monosynaptic input to superficial and deep layer pyramidal neurons was similar, and monosynaptic

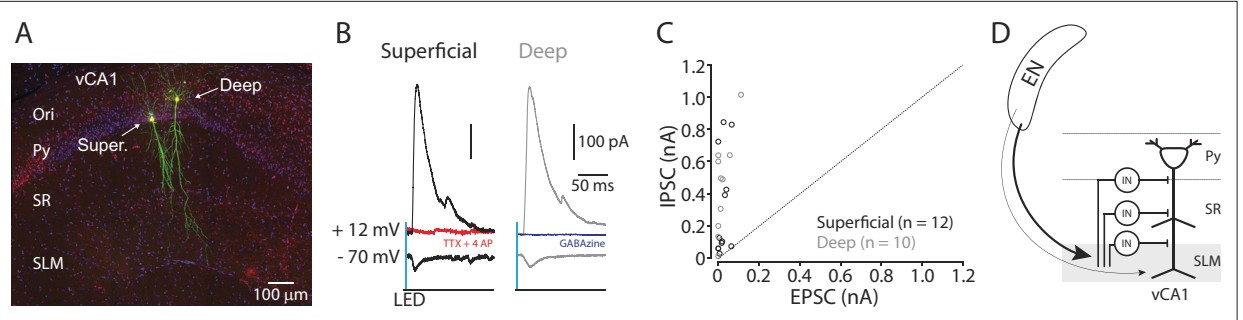

**Figure 5.** EN disynaptically inhibits ventral CA1 (vCA1) pyramidal neurons. (**A**) A confocal image of recorded pyramidal neurons in vCA1. Recorded pyramidal neurons were filled with biocytin and post-labeled with Alexa488. Ori.: oriens, Super.: superficial layer pyramidal neuron. Deep: deep layer pyramidal neuron. (**B**) Example traces of photo-evoked EPSCs and IPSCs in superficial (black trace) and deep (gray trace) pyramidal neurons. Red trace: after TTX and 4AP. Blue trace: after gabazine. (**C**) Scatter plots of peak amplitudes of EPSCs and IPSCs for each neuron. Dashed line indicates a unitary line. Superficial neurons: empty Black circles. Deep neurons: Gray circles. N=4 mice. (**D**) A summary diagram of EN→vCA1 circuit. In vCA1, EN axons innervate GABAergic neurons (IN), which in turn inhibits pyramidal neurons (both superficial and deep).

The online version of this article includes the following source data and figure supplement(s) for figure 5:

**Source data 1.** All EPSC and IPSC amplitude in *Figure 5C*.

**Figure supplement 1.** Endopiriform (EN)-mediated excitatory inputs to ventral hippocampal neurons.

input to pyramidal neurons (superficial and deep pooled) and GABAergic neurons was also comparable (*Figure 5—figure supplement 1*).

Since EN axons innervated multiple GABAergic neurons across different layers, we hypothesized they may exert stronger inhibition onto pyramidal neurons through feed-forward inhibition. To test this, we recorded EPSC and inhibitory postsynaptic current (IPSC) from superficial and deep pyramidal neurons using cesium-based internal solution in the brain slices expressing ChR2 in the EN axons (**Methods**). At the command potential of –70 mV, photo-stimulation of EN axons evoked EPSCs in the majority of pyramidal neurons in both layers (7 out of 10 neurons in superficial and 5 out of 10 neurons in deep) (*Figure 5B*). At the command potential of +12 mV, the same stimulation evoked outward postsynaptic current in the same neurons, including those without EPSCs (10 out of 10 for superficial and deep) (*Figure 5B*). This outward current was abolished by the application of TTX and 4-AP or gabazine (10 μM), indicating it is disynaptically driven and mediated by the inhibitory GABA$_A$ receptor (*Figure 5B*). When the relative strength of EPSCs and IPSCs were compared for each neuron, the strength of IPSCs overwhelmed the EPSCs in all cases (*Figure 5C*). This indicates EN axons disynaptically inhibit pyramidal neurons in vCA1 (*Figure 5D*).

## EN[vCA1-proj]. neurons receive inputs from the piriform cortex

We next examined afferents that may drive the EN→vCA1 circuit. To address this, retrograde CAV2-Cre was injected into vCA1 to express Cre into EN[vCA1-proj]. neurons, then applied the monosynaptic rabies tracing technique to specifically label its presynaptic neurons in wild-type mice (*Figure 6A*). Starter cells and presynaptic neurons were spatially mapped and quantified using AMaSiNe (*Figure 6B and C*, *Figure 6—figure supplement 1*). Since presynaptic neurons were sparse in the contralateral hemisphere, data from the ipsilateral hemisphere was analyzed. Moreover, we applied a minimum labeling threshold to determine the brain region for further analysis (**Methods**). We found the piriform cortex contained the highest number of presynaptic neurons followed by EN (*Figure 6D–F*). Within the piriform cortex, layer 2 neurons were the major source of afferent (*Figure 6G*). These data indicate EN[vCA1-proj]. neurons receive major input from piriform cortex and form a recurrent connection with neurons in EN (*Figure 6H*).

## The activity of EN[vCA1-proj]. neurons are correlated with the time mice spend in space

The downstream target of EN, vCA1, is implicated in social, odor, and object recognition memory (*Hunsaker et al., 2008*; *Okuyama et al., 2016*; *Kesner et al., 2011*). To determine whether and how the activity of EN neurons is related to this function, we expressed GCaMP8s in EN[vCA1-proj]. neurons

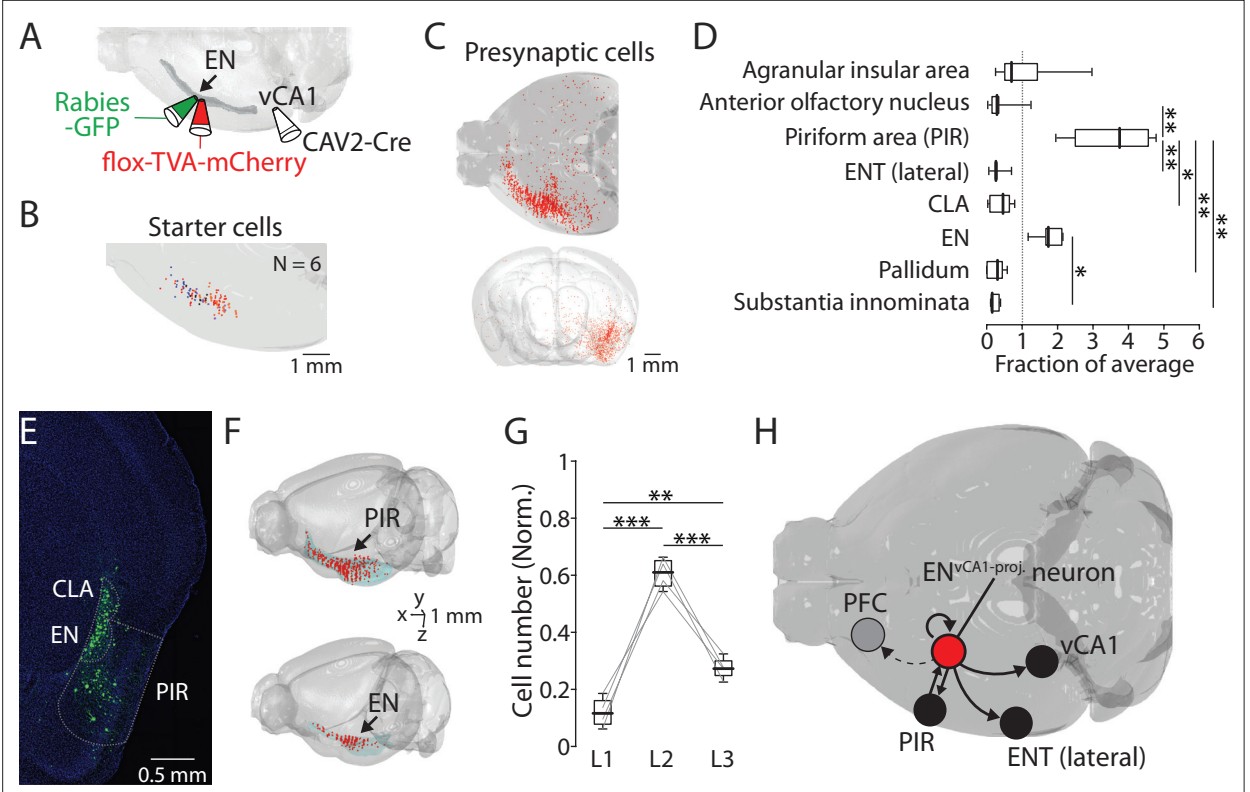

**Figure 6.** EN[vCA1-proj]. neurons receive inputs from the piriform cortex and within endopiriform (EN). (**A**) Schematic diagram of performed. After CAV2-Cre (60 nl) injection into the ventral CA1 (vCA1), flox-TVA-mCherry (100 nl) was injected into the EN. 4 wk later, Rabies-GFP (100 nl) was injected into the EN. (**B**) Location of starter cells in EN. Data from six brains were overlaid. (**C**) An example map of presynaptic cells in one brain. Top: Horizontal view. Injections were made in the left hemisphere. Bottom: Front view. (**D**) Number of presynaptic cells in different brain areas (normalized to total presynaptic cell number). Median for AIA, AON, PIR, ENT(lateral), CLA, EN, Pallidum, and SI (in fraction): 0.7, 0.3, 3.8, 0.3, 0.5, 1.7, 0.3, and 0.2. [AON, ENT(lateral), CLA, Pallidum, SI] vs PIR: p=0.016, 0.008, 0.048, 0.006, and 0.003. EN vs SI, p=0.043. one-way ANOVA with post-hoc Kruskal-Wallis test. Full list of statistical tests and p-values is in **Table 2**. (**E**) An example fluorescent image of coronal slice containing EN and PIR and labeled presynaptic cells. (**F**) Example maps of a presynaptic cells in PIR (top) and EN (bottom). (**G**) Number of presynaptic cells in PIR in different layers (normalized to the total number of labeled neurons in PIR). Median: L1 vs L2 vs L3, 0.11 vs 0.61 vs 0.27. L1 vs L2, p<0.001; L1 vs L3, p=0.005; L2 vs L3, p<0.001; one-way ANOVA, repeated comparison with Tukey-Kramer test. (**H**) Summary diagram showing an input-output circuit of EN[vCA1-proj]. neurons (red circle).

The online version of this article includes the following source data and figure supplement(s) for figure 6:

**Source data 1.** Numbers of presynaptic cells projecting to ENvCA1-proj. neurons.

**Source data 2.** Number of presynaptic piriform cells projecting to ENvCA1-proj. neurons in different layers.

**Figure supplement 1.** Presynaptic neurons of EN[vCA1-proj]. Neurons.

and monitored their activity using fiber photometry while video recording mice performing a social recognition memory tests consisting of a pretest, sociability test, and discrimination test (**Figure 7A and B**, **Methods**). The nose position of the subject mouse was tracked during tests using DeepLabCut (**Figure 7C**; **Mathis et al., 2018**; **Lauer et al., 2022**), and cumulative time and calcium events were mapped onto the arena space (**Figure 7C**, **Figure 7—figure supplement 1**). We found these values were correlated in space across all sessions (i.e. the more time the mice spent in a given space, the more the EN[vCA1-proj]. activity), including in the open field with no object or in the test arena with conspecifics or objects (**Figure 7—figure supplement 1** and **Figure 7—figure supplement 2A-B**).

Cumulative calcium events of EN[vCA1-proj]. neurons became high towards the space around objects or conspecifics when they were present and were correlated with interaction times except for sociability session (**Figure 7—figure supplement 2C-D**). In the sociability session, the cumulative calcium events around a novel conspecific and object were similar despite mice spending more time on conspecifics (**Figure 7E left**, **Figure 7—figure supplement 2C**). In contrast, in the social discrimination test, both the cumulative time and cumulative EN[vCA1-proj]. activity was higher for an

**Table 2.** ANOVA comparison of *Figure 6D*.

| Compared brain areas | | p-value |
|---|---|---|
| Agranular insular | AOB | 0.764 |
| Agranular insular | PIR | 0.617 |
| Agranular insular | ENT | 0.624 |
| Agranular insular | CLA | 0.921 |
| Agranular insular | EN | 0.971 |
| Agranular insular | Palidum | 0.588 |
| Agranular insular | Substantia innominata | 0.44 |
| AOB | PIR | 0.016 |
| AOB | ENT | 1 |
| AOB | CLA | 1 |
| AOB | EN | 0.156 |
| AOB | Palidum | 1 |
| AOB | Substantia innominata | 1 |
| PIR | ENT | 0.008 |
| PIR | CLA | 0.048 |
| PIR | EN | 0.994 |
| PIR | Palidum | 0.006 |
| PIR | Substantia innominata | 0.003 |
| ENT | CLA | 1 |
| ENT | EN | 0.091 |
| ENT | Palidum | 1 |
| ENT | Substantia innominata | 1 |
| CLA | EN | 0.317 |
| CLA | Palidum | 0.999 |
| CLA | Substantia innominata | 0.992 |
| EN | Palidum | 0.079 |
| EN | Substantia innominata | 0.043 |
| Palidum | Substantia innominata | 1 |

unfamiliar than familiar conspecific (*Figure 7D–E*, *Figure 7—figure supplement 2C*). As a result, $EN^{vCA1\text{-}proj.}$ activity and behavior had a greater degree of correlation in social discrimination than sociability.

We also tested if a similar correlation occurs in a non-social context by recording $EN^{vCA1\text{-}proj.}$ activity and behavior in the object discrimination test (*Figure 7F*). The cumulative time and calcium events were significantly correlated across different sessions i.e., familiarization, and object discrimination (*Figure 7G*, *Figure 7—figure supplement 2E*), largely consistent with the findings in the social discrimination test.

These data indicate that the activity of $EN^{vCA1\text{-}proj.}$ neurons are generally correlated with the time the mouse spends in a particular location at the basal level (i.e. open field) or task conditions. However, the degree of correlation appears to vary depending on the session, such that it was highest in the pretest/familiarization and social/object discrimination test, but lowest in the sociability test.

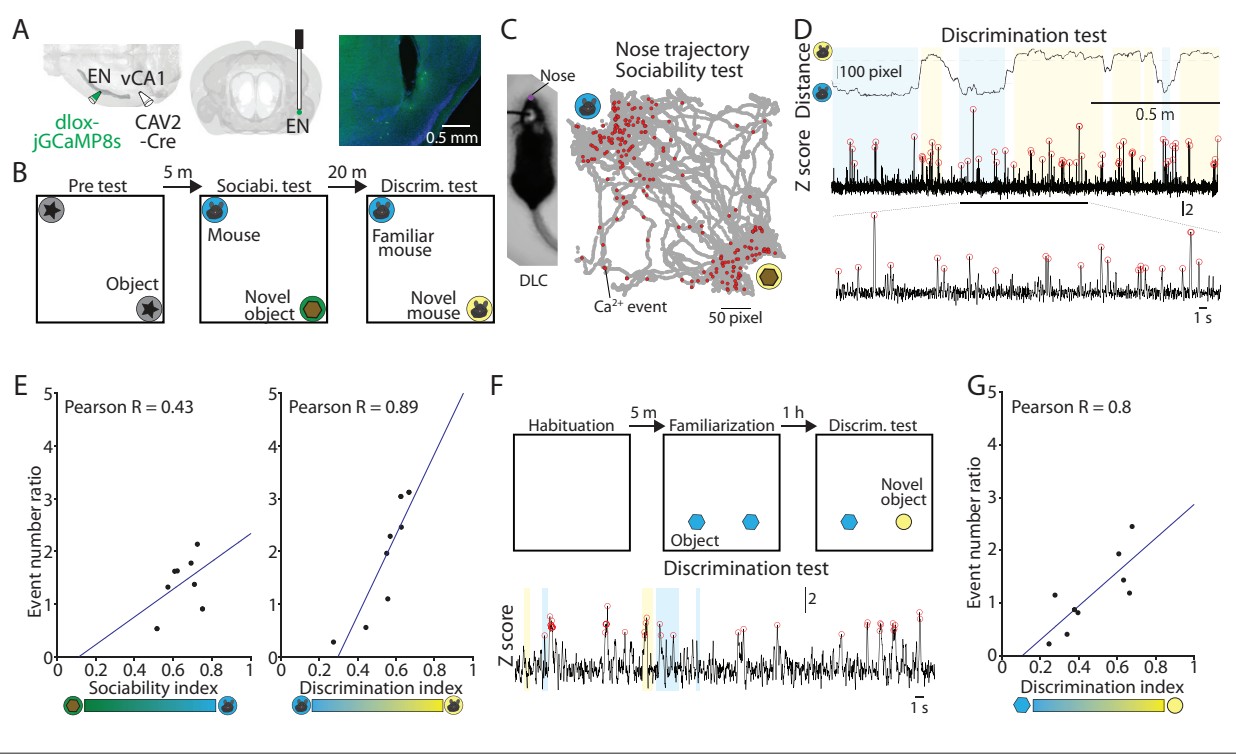

**Figure 7.** Activity of EN[vCA1-proj.] neurons correlate with social/object discrimination performance. (**A**) Right and Middle: Schematic diagrams of injection and optic cannula implantation performed. CAV2-Cre (60 nl) was injected into the vCA1, and dlox-jGCaMP8s (100 nl) was injected into the endopiriform (EN). Left: An example image showing jGCaMP8s expressing EN[vCA1-proj.] neurons and a tract of optic cannula. (**B**) The diagram of social recognition memory test. A test consisted of pretest (5 min), sociability test (10 min), and discrimination test (5 min). (**C**) Left: An example image of a mouse with a nose point marker tracked by DLC. Right: An example map of nose points and calcium events in the arena during the sociability test. (**D**) Nose distance vector (distance between subject's nose point and a center point of novel/familiar mouse chambers) and simultaneously acquired calcium signals during discrimination test. Gray dashed lines indicate the border of interaction zones. Open red circles indicate the calcium event detected (threshold: Z>2.58). A part of the calcium signal trace was expanded at the bottom for clarity. (**E**) Left: Correlation between calcium event ratios (calcium event number in the mouse interaction zone/ calcium event number in the object interaction zone) and sociability index. p=0.29, N=8 mice. Right: same as in the left panel, but for calcium event ratios (calcium event number in the unfamiliar mouse interaction zone/calcium event number in the familiar mouse interaction zone) and discrimination index. p=0.003, N=8. (**F**) The diagram of novel object recognition memory test. A test consisted of habituation (5 min), familiarization (10 min), and discrimination test (5 min). (**G**) Same as in (**E**), but for calcium event number ratios calcium event number in the unfamiliar object interaction zone/calcium event number in the familiar object interaction zone and discrimination index. p=0.01. N=9 mice.

The online version of this article includes the following source data and figure supplement(s) for figure 7:

**Source data 1.** Raw data of *Figure 7E, G*.

**Figure supplement 1.** Nose point location and calcium signal data during social memory test.

**Figure supplement 2.** Calcium signals during exploratory behavior in an open field, pretest, and familiarization test.

## Inhibition of EN[vCA1-proj.] neurons impair social/object recognition memory

Although EN[vCA1-proj.] activity was correlated with the behavior in the pretest/familiarization and discrimination test better than the sociability test, their causality is unclear. To address this, we inhibited the activity of EN[vCA1-proj.] neurons using inhibitory DREADD (hM4Di) during social and object discrimination tests (*Figure 8A–C and H*, *Figure 8—figure supplement 1A*; *Roth, 2016*). To specifically express hM4Di in EN[vCA1-proj.] neurons, CAV-2-Cre (60 nl) and AAV-flox-hM4Di (100 nl for each coordinate) were bilaterally injected into vCA1 and EN, respectively (*Figure 8A–B*). hM4Di was replaced with tdTomato for the control group. We made a protocol that allows within-subject comparison, such that a mouse goes through a test with clozapine-N-oxide (CNO, 1 mg/ kg) treatment and then the same test with saline treatment the next day, or vice versa (*Figure 8C and H*). In the social discrimination test, CNO treatment did not affect the pretest or sociability in the hM4Di or the control group, but specifically impaired social discrimination in the hM4Di group (*Figure 8D–G*, *Figure 8—figure supplement*

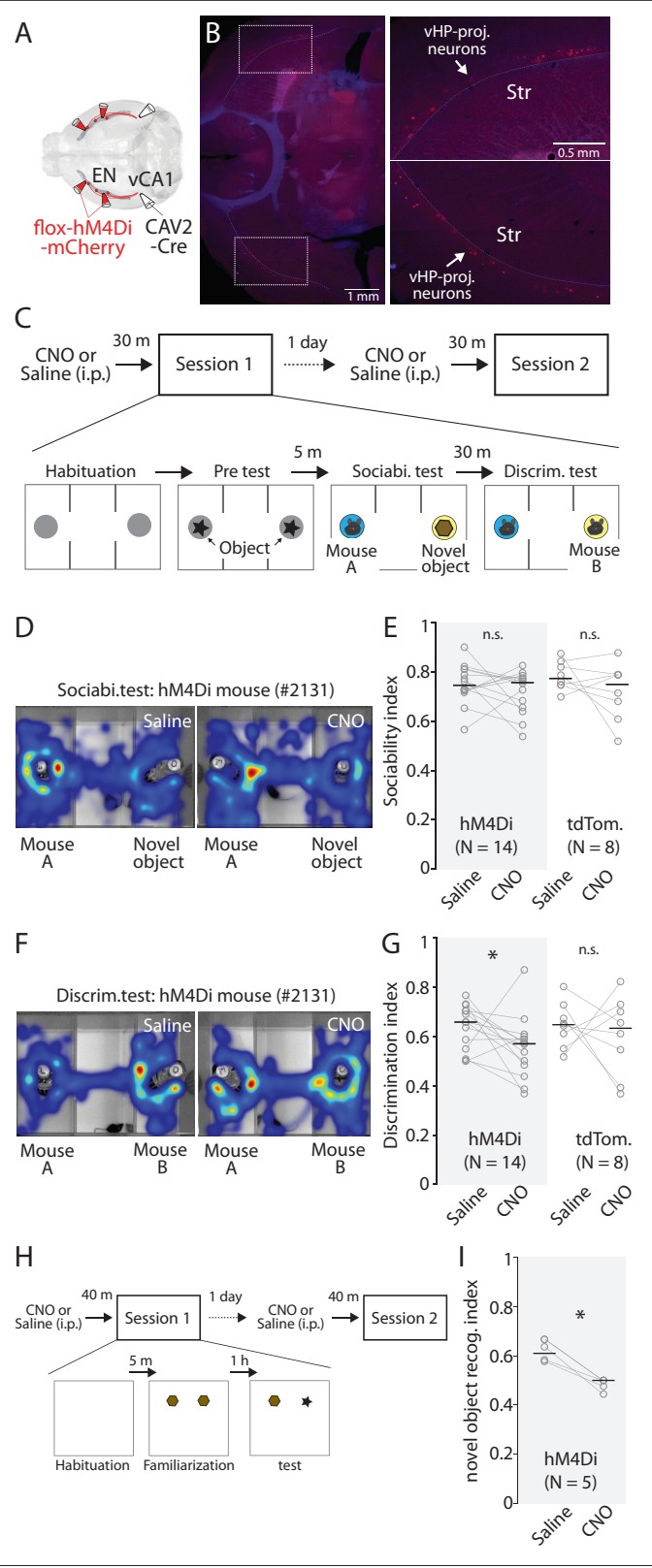

**Figure 8.** Chemogenetic inhibition of EN^vCA1-proj. neurons specifically impairs recognition memory. (**A**) Schematic diagram of injection performed. (**B**) Left: An example fluorescent image showing expression of hM4Di in EN^CA1-proj. neurons. Right: Expansion of areas marked by white rectangle in the image on the left. (**C**) The diagram of social recognition memory test. A session consisted of habituation (5 min), Pretest (5 min), Sociability test (10 min), and

*Figure 8 continued on next page*

*Figure 8 continued*

discrimination test (5 min). The group treated with a saline in session 1 was treated with clozapine-N-oxide (CNO) (1 mg/ kg) in session 2 the following day. (**D**) Example cumulative times heat map in the arena during sociability tests for the same mouse under two different treatments. (**E**) Plots comparing sociability index for hM4Di and tdTomato mice under two different treatments. Gray empty circles indicate each mouse. Gray thin lines show paired data from one mouse. Thick black bars indicate median. Median sociability index: hM4Di mice (Saline vs CNO), 0.75 vs 0.76, p=0.391; signed-rank test; tdTomato mice (Saline vs CNO), 0.77 vs 0.75, p=0.383; signed-rank test. (**F**) Example cumulative time heat maps in arena during discrimination tests for the same mouse under two different treatments. (**G**) Comparison of discrimination index for hM4Di and tdTomato mice under two different treatments. Gray empty circles indicate each mouse. Gray thin lines show paired data from one mouse. Thick black bars indicate the median. Median discrimination index: hM4Di mice (Saline vs CNO), 0.66 vs 0.57, p=0.042; signed-rank test; tdTomato mice (Saline vs CNO), 0.65 vs 0.63, p=0.742; signed-rank test. (**H**) The diagram of novel object recognition memory test. A session consisted of habituation (5 min), familiarization (10 min), and discrimination test (5 min). CNO treatment was the same as in (**C**). (**I** Same as in **G**), but for novel object discrimination index. Median discrimination index: hM4Di mice (Saline vs CNO), 0.64 vs 0.5, p=0.003; signed-rank test.

The online version of this article includes the following source data and figure supplement(s) for figure 8:

**Source data 1.** Time in each chambers/ interaction zones during sociablity test.

**Source data 2.** Time in chambers/interaction zones during discrimination test.

**Source data 3.** Interaction time with objects during novel object recognition test.

**Source data 4.** % of freezing during each test phases of trace fear conditioning test.

**Figure supplement 1.** The effect of chemoegentic inhibition of EN[vCA1-proj]. neurons on the pretest and locomotion.

**Figure supplement 2.** The effect of chemoegentic inhibition of EN[vCA1-proj]. neurons on trace fear conditioning. (**A**) Schematic of trace fear conditioning protocol. (**B**) The effect of clozapine-N-oxide (CNO) on freezing behavior during conditioning. Black indicates saline treated group and red indicates CNO treated group. Data points of individual mice are indicated by empty circles. Thick horizontal bars indicate median values. Saline vs CNO, $F_{(1, 9)}$=2.52, p=0.115; two-way ANOVA. (**C**) The effect of CNO on freezing behavior during recall to context 'A.' Median values of freezing %: Saline = 14.91, CNO = 30.13. Saline vs CNO, p=0.576; one-way ANOVA. (**D**) The effect of CNO on freezing behavior during conditional stimulus recall in context 'B.' Saline vs CNO, $F_{(1, 9)}$=0.38, p=0.54; two-way ANOVA.

1B-G). A similar effect was also observed in novel object discrimination (*Figure 8I*, *Figure 8—figure supplement 1D*).

Apart from recognition memory, vCA1 is implicated in anxiety and associative fear memory (*Jimenez et al., 2018*; *Jimenez et al., 2020*), thus, we tested the contribution of EN[vCA1-proj]. neurons to these functions. The anxiety level as determined by locomotor activity (*Prut and Belzung, 2003*) and sociability was not affected by CNO treatment in the hM4Di group (*Figure 8E*, *Figure 8—figure supplement 1E*). Similarly, fear memory as assessed by the freezing response during training (with trace fear conditioning) or recall to context or tone was not affected by CNO treatment in the hM4Di group (*Figure 8—figure supplement 2*).

Taken together, these data indicate that EN[vCA1-proj]. activity is not causally related to innate exploration behavior induced by novelty, anxiety, or fear memory. However, their activity is necessary for mice to discriminate between familiar and unfamiliar conspecific objects, suggesting their major role in general recognition memory.

## Discussion

We characterized the circuit and function of EN neurons targeting specific subregions and layers of vCA1. In contrast to other parts of the CLA complex, EN[vCA1-proj]. neurons were reciprocally connected with the piriform cortex representing a major downstream target of the olfactory bulb (*Bekkers and Suzuki, 2013*). Since social odor information is crucial for discriminating conspecifics in rodents, this circuit motif may predict the predominant role of EN[vCA1-proj]. neurons in social recognition memory, given that social odor, can engage multiple olfactory pathways innervating the piriform cortex (*Feinberg et al., 2012*). However, we found that these neurons contributed to broader domains of recognition memory. This is supported by the observations that the inhibition of EN[vCA1-proj]. activity impaired both social and non-social discrimination.

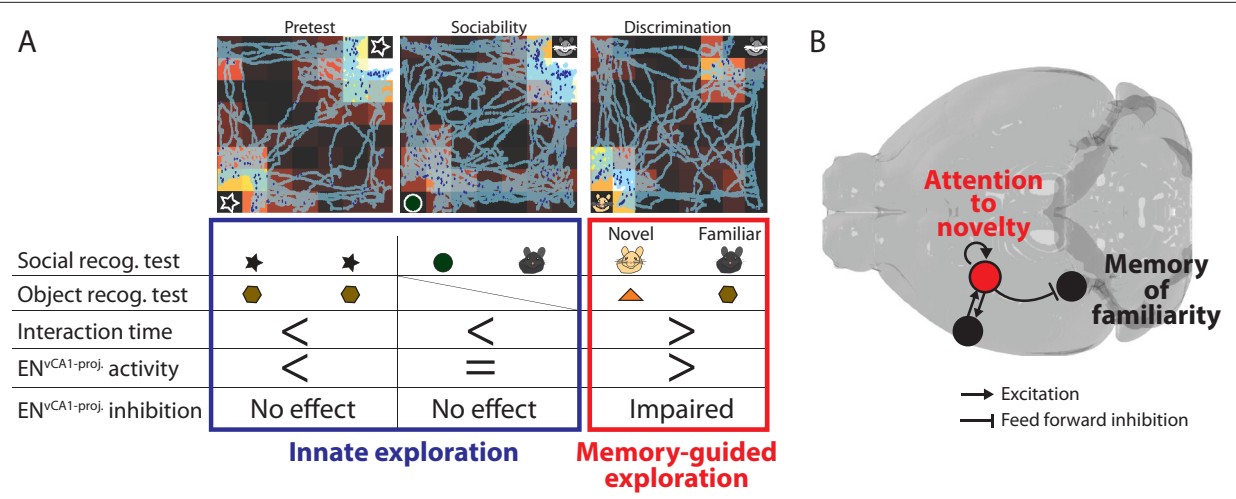

**Figure 9.** The role of EN[vCA1-proj.] neurons in recognition memory. (**A**) Summary of findings from in vivo experiments. (**B**) The model of EN[vCA1-proj.] circuit function in recognition memory. During memory-guided exploration, recurrent circuits in EN[vCA1-proj.] neurons maintain the attentional response to novel stimuli while regulating the degree of response to familiar stimuli in ventral CA1 (vCA1) via feedforward inhibition.

In all phases of recognition memory test, EN[vCA1-proj.] activity was highly correlated with the time mice spent in given locations and this correlation was biased towards novel stimuli when mice were engaged in their exploration irrespective of the stimuli being social or non-social (*Figure 9A*). Taken together, these data suggest that the function of EN[vCA1-proj.] neurons were primarily related to the detection of and attention to novel stimuli. The recurrent connections within EN, which is another prominent feature of EN[vCA1-proj.] neurons' network, potentially supporting these processes to help establish the recognition memory (*Shu et al., 2003*).

Attentional processes mediated by EN[vCA1-proj.] neurons can be considered of particular relevance for memory-guided behavior, given that inhibition of EN[vCA1-proj.] activity selectively impaired conspecific/object discrimination but not innate exploratory behavior provoked by novelty as seen in pretest, sociability, and familiarization (*Figure 9A*). The correlation of EN[vCA1-proj.] activity with novel object preference in the pretest nevertheless suggests that these neurons 'represent' the innate preference without driving it. This functional specialization, likely associated with its unique circuit connectivity to the limbic system, could differentiate EN from CLA.

Studies on EN are generally scarce, and the available evidence of EN projections to vCA1 suggests that these projections are sparse (*Behan and Haberly, 1999*). Potential reasons include the difficulty in clearly delineating the EN from CLA complex and non-uniform distribution of EN[vCA1-proj.] neurons along the antero-posterior axis. By leveraging contemporary circuit analysis tools, we demonstrated that the EN→vCA1 circuit is non-trivial; EN[vCA1-proj.] neurons were more numerous than commonly studied vCA1 afferents, including those from the septum and amygdala. Moreover, these axons exhibited a potent disynaptic inhibition of distal vCA1 pyramidal neurons, indicating their major functional implications for vCA1 activity.

Our data indicated that although EN axons terminated at the SLM of distal vCA1, they exhibited stronger connections to GABAergic neurons spanning from the Py-SR border to the SR-SLM border than to SLM neurons. The detection of EN inputs from GABAergic neurons at the Py-SR border was surprising, considering the distance between the soma position and the EN axons. However, there are several GABAergic cell types below SLM that extend their primary dendrites into SLM (*Li et al., 1992*; *Klausberger and Somogyi, 2008*), likely explaining the observed connectivity pattern. Among these are chandelier cells found at the Py-SR border, which preferentially inhibit the axon initial segments of pyramidal neurons, and are thus well suited to exert the powerful feedforward inhibition of vCA1 pyramidal neurons observed in our recordings (*Dudok et al., 2021*).

There are several potential circuit effects by which EN inputs could affect the vCA1 function. One is the induction of non-linear events in vCA1 pyramidal neurons by acting in concert with other afferents (e.g. lateral entorhinal cortex targeting SLM) (*Bilash et al., 2023*) to generate a specific output pattern in pyramidal assemblies representing task-relevant stimuli. Alternatively, EN inputs may improve the

signal to noise ratio of the information conveyed through the lateral entorhinal cortex, CA2 (in the case of social context), and possibly other afferents (e.g. CA3) (*Chiang et al., 2018*). Another scenario is tuning of vCA1 pyramidal neurons' response to familiar stimuli through a dynamic combination of a weak monosynaptic excitatory input and strong disynaptic inhibitory input. Such mechanism suggests that at a milliseconds scale novelty (EN) and familiarity (vCA1), recognition might occur though alternating rather than the simultaneous activity of brain microcircuits. Taken together, we propose a model for the role of EN$^{vCA1-proj.}$ neurons in recognition memory by balancing memory-guided attentional responses to familiarity and novelty through a combination of feedforward inhibition of vCA1 (a node for recognition of familiarity) and the recurrent circuits that contribute to sustaining attention to novelty (*Figure 9B*).

The model of promoting novelty detection by suppressing familiarity responses is consistent with previous observations showing that vCA1 pyramidal neurons predominantly respond to familiar stimuli whereas novelty response can only occur through activation of vCA1 interneurons (*Tao et al., 2022*; *Deng et al., 2019*). Nevertheless, alternating activation of EN and vCA1 could prove essential for the behavioral relevance of EN activity associated with recognition memory, without affecting innate exploratory behavior to novelty. Accordingly, the latter behavior is primarily associated with the prefrontal cortex and amygdala, areas that are only weakly targeted by EN$^{vCA1-proj.}$ neurons (*Figure 2P*, *Figure 2—figure supplement 1C*; *Wei et al., 2023*; *Franklin et al., 2017*; *Shah and Treit, 2003*).

In addition to advancing our understanding of the basic organization and function of brain circuits underlying higher cognitive processes, our findings suggest dysfunction of EN$^{vCA1-proj.}$ neurons could be a key contributing factor to the deficits in CLA complex function and recognition memory found in neuropsychiatric disorders (*Chen et al., 2023*; *Cascella et al., 2011*). Discrete EN populations can thus emerge as pathophysiological substrates, but also as important treatment targets for key symptoms of these disorders.

## Materials and methods

### Mice

All experiments were performed in accordance with standard ethical guidelines and were approved by the Danish national animal experiment committee (License number: 2021-15-0201-00801). Unless otherwise noted, all mice used were C57BL6/J mice or transgenic mice with C57BL6/J backgrounds. They were 2–4 mo old at the time of the experiment. Transgenic mice used were B6.Cg-Gt (ROSA)26Sor$^{tm14(CAG-tdTomato)Hze}$/J mice (Ai14, JAX007914) and B6J.129S6(FVB)-Slc32a1$^{tm2(cre)Lowl}$/MwarJ mice (Vgat-Cre, JAX028862). Similar number of male and female mice were used for all experiments except for behavior tests. For behavior tests, male mice were used.

### Viruses and retrograde tracers

The information on viruses used in our experiments was the following. CAV-2-Cre (PVM); AAV1-mDlx-HBB-chl-dlox-TVA_2 A_oG(rev)-dlox (v271-1, VVF); AAV1-hSyn1-dlox-TVA_2 A_mCherry_2 A_oG(rev)_dlox (v306-1, VVF); Rabies-GFP (NTNU viral core facility); Rabies-mCherry (NTNU viral core facility); AAV5-CAG-GFP (37525-AAV5, Addgene); AAV5-CAG.hChR2(H134R)-mCherry (10054-AAV5, Addgene); AAV8-hSyn-DIO-hM4Di (Gi) –mCherry (44362-AAV8, Addgene); AAV5-hEF1a-dlox-EGFP(rev)-dlox (v217-5, VVF); and retrograde AAVrg-CAG-GFP (37825-AAVrg, Addgene), AAV1-hSyn1-dlox-jGCaMP8s(rev) (v627-1, VVF). We also used retrograde traces of red Retrobeads (Lumafluor) and Cholera toxin subunit B conjugated with Alexa 647 (CTB647, Thermo Fisher).

### Stereotaxic injections

Stereotaxic injections were performed using a stereotaxic frame (Model 940, Kopf). Mice were anesthetized with isoflurane and were subcutaneously injected with buprenorphine (0.1 mg/kg) and Metacam (1.5 mg/Kg) for post-operative pain relief. After incising the scalp over the cranium, a small hole was bored with a microdrill bit, and a beveled glass pipette (Wiretrol II, 5-000-2010, Drummond Scientific Company) back-filled with mineral oil and front-filled with the material to be injected was slowly inserted into a target coordinate. After injecting a small volume (50–100 nL) with a custom-made displacement injector (based on MO-10, Narishige), the pipette was left in place for 3–5 min before slow retraction. The incision was closed with a nylon suture. The stereotaxic coordinates used

were (relative to bregma, in mm): anteroposterior (AP) –3.0; mediolateral (ML) 3.12; dorsoventral (DV) 3.7 for ventral CA1; AP +2.0; ML 0.2; DV 1.5 and 2.5 for prefrontal cortex; AP 0.0; ML 0.2; DV 1.0 and 1.5 for anterior cingulate cortex; AP 0.0; ML 1.5; DV 0.3 and 0.7 for motor cortex; and AP –5.0; ML 3.0; DV 1.5 for dorsomedial entorhinal cortex. For endopiriform injection, we injected two sites from the following coordinates: AP –0.27, ML 3.2, DV 4.2; AP 0.0, ML 3.0, DV 4.2; or AP +1.0, ML 2.7, DV 4.0. Mice were thermally supported with a feedback-controlled heating pad maintained at ~37 °C (ThermoStar Homeothermic system, RWD). Mice were used for experiments 3–5 wk post injection.

## Implantation of the optic probe

After stereotaxic injection of CAV-2-Cre into the vCA1 and AAV1-hSyn1-dlox-jGCaMP8s into the EN (AP –0.27, ML 3.2, DV 4.2), a fiber optic cannula (400 μm core diameter, 4.5 mm length, NA 0.39, R-FOC-BF-400C-39NA, RWD) was inserted towards EN. Once in the correct depth, the exposed brain surface around the implant was covered by Kwik-Cast (WPI) and was then fixed on the skull with dental adhesive resin cement (Super-Bond, SUN MEDICAL).

## Ex vivo electrophysiology

Mice were deeply anesthetized with isoflurane and decapitated. Horizontal sections (300 μm) containing ventral hippocampus were prepared by vibratome (VT1200S, Leica) in ice-cold choline solutions containing (in mM): 25 NaHCO$_3$, 1.25 NaH$_2$PO4-H$_2$O, 2.5 KCl, 0.5 CaCl$_2$, 7 MgCl$_2$, 25 D-glucose, 110 Choline chloride, 11.6 Ascorbic acid, and 3.1 C$_3$H$_3$NaO$_3$. Slices were subsequently incubated in artificial cerebrospinal fluid (ACSF) containing (in mM): 125 NaCl, 25 NaHCO$_3$, 1.25 NaH$_2$PO4-H2O, 2.5 KCl, 11 D-glucose, 2 CaCl$_2$, and 1 MgCl at 34 °C for 30 min then at room temperature (~20 °C) for at least 1 hr before the recording.

Whole-cell recording was performed with an upright microscope (BX51WI, Olympus) equipped with a motor-controlled stage and focus (MP285A and MPC-200, Sutter instrument), differential interference contrast, coolLED (model pE300), and a monochrome camera (Moment, Teledyne). Neurons were visualized with a 60 x lens (1.00 NA, LUMPlanFL N, Olympus) with the software Micro-Manager-2.0 gamma (US National Institutes of Health). Pipettes (4–5 MΩ) were pulled from thick-walled borosilicate capillary glass with a puller (model P-1000, Sutter instrument). For a voltage-clamp recording, the internal solution contained (in mM): 135 CsMeSO$_3$, 10 HEPES, 4 Mg-ATP, 0.3 Na-GTP, 8 Na$_2$-Phosphocreatine, 3.3 QX-314 (pH was adjusted to 7.35 with CsOH). In some experiments, a biocytin (4 mg/mL) and Alexa488/586 (50 μM) were further added for morphological studies. Recordings and hardware control were performed with Wavesurfer (Janelia Farm). Signals were amplified and Bessel filtered at 4 kHz with an amplifier (MultiClamp 700B, Molecular Device), then sampled at 10 kHz with a data acquisition board (USB-6343, National Instrument). Recording with access resistance change of >20% from the baseline (~30 MΩ or less) were discarded. Liquid junction potential was not corrected. All recordings were performed at ~34 °C maintained with the inline heating system (TC-324C, Warner instrument).

To stimulate channelrhodopsin-2, a blue LED was delivered at a short pulse (5 ms) through a 4 x objective lens (0.16 NA, UPlanSApo, Olympus). The light power (measured at the focal plane) was 20 mW for the recordings in normal ACSF and 40 mW for the recordings in the presence of tetrodotoxin and 4-aminopyridine. Photo-stimulation was repeated 3–6 times (at intervals of 10 or 20 sec) to obtain an average response trace. Photo-evoked responses less than 2 standard deviations of the baseline were considered as a 'zero' response. The mean amplitudes indicate mean amplitudes between LED stimulus-onset to 50 ms of average responses.

Drugs used for ex vivo electrophysiology were purchased from Merck, TOCRIS, and Hellobio.

## Quantification of presynaptic neurons

Mice were deeply anaesthetized with a ketamine (120 mg/kg) and xylazine (24 mg/kg) mixture and were transcardially perfused with 4% paraformaldehyde (PFA). The brains were extracted and immersed in 4% PFA overnight followed by 20 and 30% sucrose solution for a cryoprotection. The brains were subsequently embedded in O.C.T compound (Tissue TEK, Sakura) in a mold and frozen on the dry ice. Coronal slices (100 μm) of the entire brain were cut with Cryostat (CM3050S, Leica) and were washed with phosphate-buffered saline (PBS) before DAPI staining. All sections were mounted

onto glass slides with a coverslip (thickness:0.13–0.17 mm, Hounisen) and mounting media (DAKO, S3023, Agilent Technologies).

The injection sites and their specificity were verified by registering the images into Allen Brain atlas using QuickNII and VisuAlign (*Puchades et al., 2019*).

For the visualization and quantification of presynaptic neuron distribution in a 3D mouse brain, epifluorescence images were acquired using a Slidescanner (4231x3,462 pixels, 10 x objective, Olympus VS120) and analyzed with the MATLAB-based program AMaSiNe (*Song et al., 2020*). Specifically, for *Figure 1*, coronal sections between AP +3 mm and –4 mm were registered in AMaSiNe, which semi-automatically detects fluorescently labeled presynaptic cells across all sections. To control for transfection variability (which may arise from slight differences in injection volume or site), we normalized the number of presynaptic neurons in the specified brain areas (shown in *Figure 1H*) by the total number of presynaptic neurons in those regions.

For the monosynaptic rabies tracing study in *Figure 6*, the brain areas containing over 1% presynaptic cells of total presynaptic cells have proceeded with further analysis. Fraction of presynaptic cells in the different areas was calculated by dividing the number of presynaptic cells in each brain areas by the average number of presynaptic neurons in all brain areas.

For the study in *Figure 2*, the sections were immunostained with NeuN before imaging. Here, sections were immersed with 5% normal goat serum and 0.2% Triton X for 30 min at room temperature before incubation with NeuN primary antibody (anti-rabbit, 1:1000) (ABN78, Milipore) at 4 °C for overnight. The sections were then washed with PBS and were further incubated with a secondary antibody (goat anti-rabbit conjugated with Alexa 488 or 568, 1:500) (A11008 or A11036, Thermo Fisher) for 1 hr at room temperature before mounting. A single-plane image of CLA complex was acquired with a confocal microscope (10 x objective, 1024×1024 pixels, Zeiss). Two to four images, range of AP: +1.4 to –0.2, were collected per brain. Fraction of double-labelled neurons (with different retrograde tracers) was calculated for each brain by dividing the total number of double-labeled neurons by the total number of vCA1-projecting neurons (labeled with one color).

## Quantification of fluorescence signal from axons

For *Figure 3* and *Figure 3—figure supplement 1*, horizontal section (100 µm) images were taken with Sliderscanner (10 x lens, pixel: 2560×3072). All images from the same brain were taken with the same light intensity and detection setting. After background subtraction, GFP signals in the ventral, intermediate, and dorsal CA1 columns were normalized by the highest signal in the ventral CA1 in the same brain.

For *Figure 2*, coronal sections (100 µm) were immunostained with GFP antibody before imaging. Here, sections were washed with PBS, the sections were incubated in a blocking solution containing 5% normal goat serum and 0.2% Triton X for 30 mins at room temperature before incubation with GFP primary antibody (anti-rabbit, 1:1000) (NB600-308, MNovus Biologicals) in blocking solution at 4 °C for overnight. The sections were then washed with PBS and were further incubated with a secondary antibody (goat anti-rabbit conjugated with Alexa 488, 1:300) (A11008, Thermo Fisher) for 1 hr at room temperature before mounting. Images were taken with Slidescanner (10 x, pixel: 3072×3584). Signal per pixel in ROI was calculated for each projection site after background subtraction.

## Social memory test with chemogenetic inhibition of EN$^{vCA1-proj}$. neurons

Social memory test is based on previous articles with slight modifications (*Mesic et al., 2015*; *Rein et al., 2020*). All tested mice were habituated to the handling for 3 d. The water-soluble CNO purchased from Hellobio was dissolved in saline (final concentration was 0.1 mg/ml). Prior to the test day, we divided the subject mice into two groups. On the first test day (session 1), one group received clozapine-N-oxide (CNO, 1 mg/kg, i.p.) and another group received a vehicle. After 30 mins, the subject mice were placed in a three-chamber box for 5 min (habituation), then in the same box with two pencil chambers containing the same object in opposite corners for 5 min (pretest). After 5 min break in their home cage, the subject mice were again placed in the same box, but this time, an object in one pencil chamber was replaced with a different object, and an object in another pencil chamber was replaced with a conspecific (3–4 wk old male, habituated to the handling and pencil chamber for 3 d prior to the session 1). This 'sociability test' lasted for 10 min and was followed by a social discrimination test after 30 min break in the home cage. The social discrimination test lasted for 5 min and

consisted of the same setting as the sociability test except an object in a pencil chamber, which was replaced with a new conspecific (i.e., novel mice). On the following day, CNO and saline treatments were swapped between the groups, and all mice went through the same series of tests (session 2) for the within-subject comparison of the CNO effect. The position of the pencil chamber including an object or a conspecific was counterbalanced. The tests were performed under 6–10 lx room light and video was recorded at 25 frames/s with a GigE camera (Basler ace acA1300-60gm, Basler) placed at the ceiling.

### Trace fear conditioning

After 1 wk of the social memory test, subject mice were examined for their tone-associated fear memory and extinction. One group received CNO while another group received a vehicle 45 min before the test. On conditioning day, mice were placed in a box with a metal grid floor and ethanol odor (context A) and after 3 min were delivered with a 20 s pure tone (2 kHz, 20 dB) followed by 18 s time gap (trace) and 2 s footshock (0.5 mA). The exposure to the same tone-trace-shock sequence was repeated two more times with an inter-trial interval of 2 min. In the following day, mice were re-exposed to the context A for 3 min to assess their fear memory to the context. From day 3, mice were placed in a new context, the context B (smooth floor, breach odor), and went through the same protocol as in conditioning day but without shocks. The process was repeated over 5 d to assess the extinction of the fear memory to the tone and trace. All test was performed in the dark (but the infrared light was on), and behavior was video recorded at 25 frames/s with an infrared GigE camera (Basler ace acA1300-60gm, Basler).

### Novel object recognition test

The novel object recognition test is based on the previous article with slight modifications (*Mesic et al., 2015*). The subject mice were divided into two groups and CNO treatment was done as same as social memory test. 40 min after CNO/vehicle i.p. injection, the subject mice were placed in an open field box (50×50 cm) for 5 min (habituation). In the following familiarization session, two same shape objects are placed in the open field box, and then the subject mice explored for 10 min. After 1 hr break in their home cage, the subject mice are were again placed in the same box, but one object was replaced by a novel object. The subject mice are allowed to explore the box for 5 min in a discrimination session.

### Social memory test and novel object recognition test with in vivo fiber photometry

Optic probe-implanted mice were habituated handling for 3 d prior to the test. On the test day, the subject mouse attached with the patch code was placed in an open field box (50×50 cm) for 5 min (habituation). After habituation, the mouse was placed in a home cage for resting and waiting following session. For the pretest, pencil chambers containing the same objects were placed at the corners of the open field box. The pretest was tested for 5 min. For the sociability test, a novel mouse (4 wk old, male) or a novel object was placed in pencil chambers. Sociability test took 10 min. After the sociability test, the subject mouse returned to a home cage and rest for 20 min. After 20 min rested, the subject mouse was tested discrimination session for 5 min. In the discrimination test, a novel mouse (4 wk old, male) or a familiar mouse that was used as the stimulus mouse in the sociability test were placed in pencil chambers. Both in the sociability and discrimination test, the positions of pencil chambers were counterbalanced between subject mice. The tests were performed under 6–10 lux room light and video was recorded at 60 frames/s with a GigE camera (Basler ace acA1300-60gm, Basler) placed at the ceiling. Video acquisitions were controlled by Bonsai. Simultaneous start of video recording and calcium signal recording was controlled by Raspberry Pi.

After 1 wk of the social memory test, subject mice were examined for a novel object recognition test. The procedure, the arenas, and objects were used the same as in the novel object recognition test with chemogenetic inhibition of $EN^{vCA1-proj}$. neurons.

### Behavior data acquisition and analysis

Ethovision (version 16 or 17, Noldus) was used for the hardware control, and data acquisition, and initial quantification of the behavior. Behavioral variables, including movement trajectory, times spent

in predefined areas, total distance traveled, travelling velocity, and freezing were extracted and further analyzed/nested with custom functions in the Matlab (R2020b). For the social memory test, interaction zones were defined as an area between the edge of the pencil chamber and 3 cm away from the edge. When the nose of the subject mouse was in the zone, the time was counted as interaction time. Sociability index was calculated by dividing the total interaction time with mouse by the total interaction time between a mouse and the object. The discrimination index was calculated by dividing total interaction time with a novel mouse by the total interaction time with a familiar mouse and a novel mouse. Data from mice were excluded if the discrimination index was less than 0.5 during saline treatment. For the trace fear conditioning test, mice were considered 'freezing' 'if pixel change between frames was less than 0.1% for more than 1 s.

Deeplabcut (ver.2.3) was used for tracking of annotated body points of the subject mice connected with patch code for calcium imaging of EN$^{vCA1-proj}$. neurons during social memory tests and novel object recognition tests. Nose points were used for further analysis using Matlab (R2020b). To define the interaction zone, reference points were set at the center of the pencil chamber. The areas that distance between nose point and the reference point was less than 100 pixels for the social memory test and 50 pixels for the novel object recognition test were dedicated to the interaction zone.

### Fiber photometry and analysis

Population activity of EN$^{vCA1-proj}$. neurons were recorded using fiber photometry (Doric Lenses). GCaMP8s was excited with 470 nm light at 20–30 µW measured at the tip (the isosbestic point used was 415 nm (15 µW)). Calcium-dependent and independent signals were collected using lock-in mode. Signals were detected at 10 x gain. Recorded signals were Butterworth filtered (low-pass at 40 Hz) and calcium signals (in z-score) were extracted using Doric neuroscience studio V6 (Doric Lenses). Signals were further down-sampled to 60 Hz using a Matlab (R2020b) signal analyzer to match the behavioral sampling data (60 fps). Z-score >2.56 (alpha = 0.01) was determined to be a calcium event. For the heat map of cumulative time and calcium events, spatial binning was 50 pixels. To make a correlation map (*Figure 7—figure supplement 1* and *Figure 7—figure supplement 2A*), correlations between the column of cumulative time heat map and cumulative calcium event heat map were tested using the Matlab corr function.

### Statistical analysis

Data and statistical analysis were performed with MATLAB R2020b (Mathworks). Unless otherwise noted, group data was presented as median, and first and third quartiles are shown for dispersion. Statistics were performed using a non-parametric test (signed-rank test for paired data and rank-sum test for unpaired data). One-way ANOVA (post hoc test: Tukey-Kramer test or Kruskal-Wallis test) was applied for multi-group comparison. Two-way ANOVA was performed for comparison of multi-factors. Pearson correlation coefficient was applied for correlation analysis. Significance was defined as *$p<0.05$, **$p<0.01$, ***$p<0.001$. The outlier was detected by the Grubbe test (GraphPad, significant level: 0.05). Sample numbers were indicated as 'n,' and animal numbers were indicated as 'N'.

## Acknowledgements

We thank Wen-Hsien Hou, Chihiro Nakamoto, Ana Cicvaric, Hui Zhang, and Elizabeth Wood for constructive discussion of this study. We thank Peter Bjerge, Bjark B Brix, and Dennis Olesen for technical help. We acknowledge the bioimaging core facility, Health, Aarhus University, Denmark, for the use of equipment and support.

# Additional information

## Funding

| Funder | Grant reference number | Author |
|---|---|---|
| Lundbeck Foundation | | Jelena Radulovic<br>Naoki Yamawaki<br>Asami Tanimura |

The funders had no role in study design, data collection and interpretation, or the decision to submit the work for publication.

## Author contributions

Naoki Yamawaki, Conceptualization, Data curation, Investigation, Methodology, Writing - original draft, Writing - review and editing; Hande Login, Investigation, Methodology, Project administration; Solbjørg Østergaard Feld-Jakobsen, Data curation, Formal analysis, Investigation, Methodology; Bernadett Mercedesz Molnar, Data curation, Software, Formal analysis, Investigation, Methodology; Mads Zippor Kirkegaard, Aleksandra Okrasa, Data curation, Formal analysis; Maria Moltesen, Data curation; Jelena Radulovic, Supervision, Writing - original draft, Project administration, Writing - review and editing; Asami Tanimura, Conceptualization, Data curation, Formal analysis, Supervision, Validation, Investigation, Visualization, Methodology, Writing - original draft, Project administration, Writing - review and editing

## Author ORCIDs

Naoki Yamawaki ⓘ https://orcid.org/0000-0001-8253-2059
Asami Tanimura ⓘ https://orcid.org/0000-0003-4209-9793

## Ethics

All experiments were performed in accordance with standard ethical guidelines and were approved by the Danish national animal experiment committee (License number: 2021-15-0201-311 00801).

Reviewer #1 (Public review): https://doi.org/10.7554/eLife.99642.4.sa1
Reviewer #2 (Public review): https://doi.org/10.7554/eLife.99642.4.sa2
Author response https://doi.org/10.7554/eLife.99642.4.sa3

# Additional files

## Supplementary files

Supplementary file 1. p-values of correlation plots in *Figure 7—figure supplement 1*.

Supplementary file 2. p-values of correlation plots in *Figure 7—figure supplement 2A*.

MDAR checklist

## Data availability

This manuscript does not contain newly generated source data and codes. Figure 1—source datas 1 and 2, Figure 2—source data 1, Figure 3—source data 1, Figure 4—source datas 1 and 2, Figure 5—source data 1, Figure 6—source datas 1 and 2, Figure 7—source data 1, Figure 8—source datas 1–4 contain the raw data used to generate the figures.

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
