## [Editor Report · eLife Assessment]

This **important** study offers insights into the function and connectivity patterns of a relatively unknown afferent input from the endopiriform to the CA1 subfield of the ventral hippocampus, suggesting a neural mechanism that suppresses the processing of familiar stimuli in favor of detecting memory guided novelty. The strength of evidence is **convincing**, with careful anatomical and electrophysiological circuit characterization. The work will be of broad interest to researchers studying the neural circuitry of behavior.

---

## [Referee Report · Reviewer #1 (Public review)]

Summary:

The anatomical connectivity of the claustrum and the role of its output projections has, thus far, not been studied in detail. The aim of this study was to map the outputs of the endopiriform (EN) region of the claustrum complex, and understand their functional role. Here the authors have combined sophisticated intersectional viral tracing techniques, and ex vivo electrophysiology to map the neural circuitry of EN outputs to vCA1, and shown that optogenetic inhibition of the EN→vCA1 projection impairs both social and object recognition memory. Interestingly the authors find that the EN neurons target inhibitory interneurons providing a mechanism for feedforward inhibition of vCA1.

Strengths:

The strength of this study was the application of a multilevel analysis approach combining a number of state-of-the-art techniques to dissect the contribution of the EN→vCA1 to memory function.

In addition the authors conducted behavioural analysis of locomotor activity, anxiety and fear memory, and complemented the analysis of discrimination with more detailed description of the patterns of exploratory behaviour.

---

## [Referee Report · Reviewer #2 (Public review)]

Summary:

Yamawaki et al., conducted a series of neuroanatomical tracing and whole cell recording experiments to elucidate and characterise a relatively unknown pathway between the endopiriform (EN) and CA1 of the ventral hippocampus (vCA1) and to assess its functional role in social and object recognition using fibre photometry and dual vector chemogenetics. The main findings were that the EN sends robust projections to the vCA1 that collateralise to the prefrontal cortex, lateral entorhinal cortex and piriform cortex, and these EN projection neurons terminate in the stratum lacunosum-moleculare (SLM) layer of distal vCA1, synapsing onto GABAergic neurons that span across the Pyramidal-Stratum Radiatum (SR) and SR-SML borders. It was also demonstrated that EN input disynaptically inhibits vCA1 pyramidal neurons. vCA1 projecting EN neurons receive afferent input from piriform cortex, and from within EN. Finally, fibre photometry experiments revealed that vCA1 projecting EN neurons are most active when mice explore novel objects or conspecifics, and pathway-specific chemogenetic inhibition led to an impairment in the ability to discriminate between novel vs. familiar objects and conspecifics.

This is an interesting mechanistic study that provides valuable insights into the function and connectivity patterns of afferent input from the endopiriform to the CA1 subfield of the ventral hippocampus. The authors propose that the EN input to the vCA1 interneurons provides a feedforward inhibition mechanism by which memory-based novelty detection could be promoted. The experiments are carefully conducted, and the methodological approaches used are sound. The conclusions of the paper are supported by the data presented.

---

## [Author Response]

The following is the authors’ response to the previous reviews.

**Reviewer #2 (Public review):**
Summary:Yamawaki et al., conducted a series of neuroanatomical tracing and whole cell recording experiments to elucidate and characterise a relatively unknown pathway between the endopiriform (EN) and CA1 of the ventral hippocampus (vCA1) and to assess its functional role in social and object recognition using fibre photometry and dual vector chemogenetics. The main findings were that the EN sends robust projections to the vCA1 that collateralise to the prefrontal cortex, lateral entorhinal cortex and piriform cortex, and these EN projection neurons terminate in the stratum lacunosum-moleculare (SLM) layer of distal vCA1, synapsing onto GABAergic neurons that span across the Pyramidal-Stratum Radiatum (SR) and SR-SML borders. It was also demonstrated that EN input disynaptically inhibits vCA1 pyramidal neurons. vCA1 projecting EN neurons receive afferent input from piriform cortex, and from within EN. Finally, fibre photometry experiments revealed that vCA1 projecting EN neurons are most active when mice explore novel objects or conspecifics, and pathway-specific chemogenetic inhibition led to an impairment in the ability to discriminate between novel vs. familiar objects and conspecifics.The authors have addressed most of my concerns, but a few weaknesses remain :(1) I expected to see the addition of raw interaction times with objects and conspecifics for each phase of social testing (pre-test, sociability test, social discrimination), as per my comment on including raw data. However, the authors only provided total distance traveled and velocity, and total interaction time in Figure S9, which is less informative.

We apologies for missing the request. We have added the raw interaction times in Fig. S9G.

(2) The authors observed increased activity in vCA1-projecting EN neurons tracking with the preferred object during the pre-test (object-object exploration) phase of the social tests, and the summary schematic (Figure 9A) depicts animals as showing a preference for one object over the other (although they are identical) in both the social and object recognition tests. However, in the chemogenetic experiment, the data (Fig S9B) indicate that animals did not show this preference for one object over another, making the expected baseline for this task unclear. This also raises an important question of whether the lack of effect from chemogenetic inhibition of vCA1-projecting EN neurons could be attributed to the absence of this baseline preference.

We appreciate the comments. In Fig. S9B, although the group median at baseline (pretest) showed no preference for one object, individual subjects displayed a preference for one object (i.e., each data point deviated positively or negatively from 0.5) in saline condition. Therefore, we do not think that a lack of baseline preference accounts for the absence of the inhibition effect in the pretest.

Additionally, the finding that vCA1-projecting EN activity is associated with the preferred object exploration appears to counter the authors' argument that novelty engages this circuit (since both objects are novel in this instance). This discrepancy warrants further discussion.

This is an interesting point. One possibility is that during the pretest, EN activity simply "reports" or "represents" the interaction time without driving exploratory preference. This aligns with our DREADD experiment data, which show that inhibition of EN neurons produced no overall behavioral effect. Innate exploratory behavior has been attributed to various circuits, including the medial preoptic area → PAG circuit (Ryoo et al., 2021, Front. Neuro.) and the Septal → VTA circuit (Mocellin et al., 2024, Neuron). We found no direct projection from these areas to EN (Fig. 6), but such connections could be established di- or polysynaptically. Moreover, these circuits could be driven by common inputs, such as the locus coeruleus or the cholinergic system for arousal, with only specific downstream targets, excluding EN, playing a key role in driving innate exploration and preference.

We have inserted the following sentence in discussion (line 253-255):

“The correlation of ENvCA1-proj. activity with novel object preference in the pretest nevertheless suggests that these neurons 'represent' the innate preference without driving it.”

**Recommendations for the authors:**

**Reviewer #2 (Recommendations for the authors):**
Line 209: Please remove the reference to neural activity 'predicting' behavior, as correlation analysis does not imply predictive power.

We now have changed the phrase to “Although EN^vCA1-proj.^ activity was correlated with the behavior…”

Line 236: It is unclear what is meant by: 'This circuit motif may predict the predominant role of ENvCA1-proj. neurons in social recognition memory'

We have changed the sentence to the following for the clarity:

“Since social odor information is crucial for discriminating conspecifics in rodents, this circuit motif may predict the predominant role of ENvCA1-proj. neurons in social recognition memory, given that social odor can engage multiple olfactory pathways innervating the piriform cortex.”

Fig 7 title: insert 'with' after correlates: 'Activity of ENvCA1-proj. neurons correlates social/object discrimination performance'

Corrected.

Fig S1 title: 'Projecing' typo.

Corrected.

Fig S8: Please rephrase for clarity: 'In pretest, the object was aligned by longer interaction time (preferred object is plotted in right side)'

We now have rephrased the sentence to:

“In the pretest plot, the object that the mice interacted with more is placed on the right side.”

References:

A septal-ventral tegmental area circuit drives exploratory behavior. Mocellin, Petra et al. Neuron, Volume 112, Issue 6, 1020-1032.e7

An inhibitory medial preoptic circuit mediates innate exploration. Ryoo, Jia et al. Front. Neurosci., 23 August 221. Volume 15- 2021